# Obesity and Bone: A Complex Relationship

**DOI:** 10.3390/ijms222413662

**Published:** 2021-12-20

**Authors:** Giuseppe Rinonapoli, Valerio Pace, Carmelinda Ruggiero, Paolo Ceccarini, Michele Bisaccia, Luigi Meccariello, Auro Caraffa

**Affiliations:** 1Orthopaedic and Traumatology Unit, Department of Medicine, University of Perugia, 06156 Perugia, Italy; valeriopace@doctors.org.uk (V.P.); paoloceccarini84@gmail.com (P.C.); auro.caraffa@unipg.it (A.C.); 2Orthogeriatric Service, Geriatric Unit, Institute of Gerontology and Geriatrics, Department of Medicine, University of Perugia, 06156 Perugia, Italy; carmelinda.ruggiero@unipg.it; 3Department of Orthopaedics and Traumatology, AORN San Pio “Gaetano Rummo Hospital”, Via R.Delcogliano, 82100 Benevento, Italy; michelebisa@yahoo.it (M.B.); drlordmec@gmail.com (L.M.)

**Keywords:** obesity, osteoporosis, fracture, bone fragility, obese fracture site paradox

## Abstract

There is a large literature on the relationship between obesity and bone. What we can conclude from this review is that the increase in body weight causes an increase in BMD, both for a mechanical effect and for the greater amount of estrogens present in the adipose tissue. Nevertheless, despite an apparent strengthening of the bone witnessed by the increased BMD, the risk of fracture is higher. The greater risk of fracture in the obese subject is due to various factors, which are carefully analyzed by the Authors. These factors can be divided into metabolic factors and increased risk of falls. Fractures have an atypical distribution in the obese, with a lower incidence of typical osteoporotic fractures, such as those of hip, spine and wrist, and an increase in fractures of the ankle, upper leg, and humerus. In children, the distribution is different, but it is not the same in obese and normal-weight children. Specifically, the fractures of the lower limb are much more frequent in obese children. Sarcopenic obesity plays an important role. The authors also review the available literature regarding the effects of high-fat diet, weight loss and bariatric surgery.

## 1. Introduction

Obesity represents a growing social problem, which leads to significant efforts by healthcare systems all over the world to avoid the serious consequences that it can have on health.

Obesity, according to the World Health Organization’s (WHO) definition, is an abnormal or excessive fat accumulation that may impair health. The BMI (Body Mass Index) is universally used as a reference measure. The WHO states that the obese state is defined as having a BMI ≥ 30.0, and reports that, in 2016, more than 1.9 billion adults, 18 years and older, were overweight (BMI ≥ 25). Of these, over 650 million were obese and, worldwide, obesity has nearly tripled since 1975 [1]. It is predicted that 57.8% of the elderly population will be overweight or obese by 2030 [2].

Worrying data are also found in children, in whom overweight and obesity conditions are calculated thorough different parameters, based on age (WHO Growth Reference median) [3]. The prevalence of overweight and obesity among children and adolescents aged 5–19 has risen dramatically from 4% in 1975 to over 18% in 2016 [1].

Obesity can lead to very serious consequences on various organs and systems. More controversial are its effects on bone. We believe it is very important to define what effects obesity can cause on bone. In fact, in consideration of the fact that BMD (bone mineral density) is not sufficient, on its own, to define whether an individual is at greater risk of fracture, we have to detect other factors, independent of BMD, which can be defined as risk factors. If scientific evidence clarifies its significant role in increasing fracture risk, obesity could be included in the various algorithms that can be used to quantify fracture risk.

This narrative review aims to carefully examine the literature on the influence that obesity has on bone health, both in children and adults, trying to clarify a topic that is still debated and remains partially controversial. Although there are numerous studies and reviews published on this topic [4,5,6,7,8], this review aims to be as complete as possible, examining all the facets related to this problem, also broadening the discussion to the consequences of bariatric surgery and childhood obesity.

The risk of fracture depends on bone fragility and the propensity for falls. We will examine these two risk factors both in general and in obesity.

## 2. Bone Fragility and Assessment Methods

Bone fragility is a condition that leads to an increased risk of fracture. The disease that is characterized by bone fragility is called osteoporosis. It is caused by a deficit in bone “quantity” or a deficit in bone “quality” (or both) [9,10]. Bone quantity can be easily measured, with proved instruments as dual energy X-ray absorptiometry (DEXA) or quantitative ultrasounds (QUS). Conversely, bone quality is more difficult to evaluate with the available measurement tools. For this reason, at present, the diagnosis of osteoporosis is made with the use of the available instrumental methods, which calculate the BMD, combined with questionnaires. The most used questionnaire is the FRAX (Fracture Risk Assessment Tool), released in 2008 by the WHO. The FRAX provides for the collection of data relating to the patient, including all personal data (age, sex, race, weight, etc.) and the possible risk factors for fragility fractures [11]. In fact, there are some instrumental methods for assessing bone quality, which analyze bone structure. The use of QCT (Quantitative Computed Tomography) technology is now widespread. High-resolution peripheral quantitative computed tomography (HR-pQCT) is considered even more reliable. With the HR-pQCT, it is possible to evaluate the volumetric BMD (vBMD). It is unquestionably superior to DEXA in evaluating bone structure, and, therefore, bone quality [12,13]. In the obese, more specifically, other evaluation methods have been proposed. In a study by Ruosi et al. [14], the authors report that, in 54 obese subjects, BMD was normal or slightly below normal, but the spinal deformity index (SDI) showed vertebral body deformities in 87.5% of the patients and 10% in controls, signs of morphometric vertebral fractures. This is a tangible sign of DEXA deficiency in defining the risk of fracture in the obese. For the indirect evaluation of bone microarchitecture, the TBS (trabecular bone score) algorithm was proposed. TBS is a textural index based on pixel gray-level variations in the LS DEXA image, and it turned out to be quite reliable. In a study by Romagnoli et al. [15], TBS was found to be inversely related to BMI, suggesting that an increase in BMI has a negative impact on bone quality. Furthermore, at the 27th American Association of Clinical Endocrinologists (AACE) meeting, held on 2018, another method was presented for a more correct assessment of the risk of fracture in the obese patient [16]. It would be sufficient to calculate the LS BMD/BMI ratio. In obese subjects, the TBS and LS BMD/BMI ratios are almost always lower than the BMD assessed with DEXA [17].

Since it has been demonstrated that obese individuals have a greater risk of fracture than non-obese [18,19,20], this leads us to conclude that the latter two measurement criteria described can give us a better perspective of the risk of fracturing of obese individuals.

## 3. Obesity and Mineral Bone Density

For many years it was thought that the people most at risk of fractures were lean women. Lower BMI was thought to increase the risk of osteoporosis, while the higher body weight to give protection against fractures [21,22,23]. Several studies have been published showing a positive relationship between BMI and BMD [21,22,23,24]. In a cross-sectional study, Santos et al. assessed 128 subjects aged between 80 and 95 years and reported that obesity was shown to be a protective factor for osteopenia/osteoporosis in the spine (OR: 0.43; CI: 0.20–0.93) and femur (OR: 0.27; CI: 0.12–0.62), independently from sex [25]. The same conclusions were reported by other authors [26,27,28,29,30,31], while a few studies highlighted the positive correlation between BMI and BMD in post-menopausal women [22,23,32], and a study by Cherif et al. showed an overall high bone density in obese women [33].

During menopause, higher body weight seems to decelerate bone loss [34]. 

The higher BMD in obese subjects, reported by many authors, is attributable to the mechanical effect of body weight on bone [20,35,36,37,38]. The scientific literature shows how weight bearing increases bone density by acting also at the cellular level [39]. Studies conducted on animals show that osteocytes are particularly sensitive to biomechanical stress [40]. They die by apoptosis in the absence of loading [41], while, when the shear stress signal is picked up by the osteocytes [42], they do not undergo apoptosis, and their secretion of sclerostin is suppressed [43]. At the same time, the action of the osteoclasts is repressed, and osteoblastic differentiation is stimulated [44,45,46]. Garnero et al. found a decrease of biochemical bone markers in obese people, with greater decline of bone-resorption markers than bone formation ones [47]. This phenomenon would confirm that body weight gain leads to a positive bone balance.

The effect of mechanical loading on bone is determined more by the lean mass than by the fat mass [48,49,50]. In obese individuals, there is an increase in fat mass, but also in lean mass (excluding sarcopenic obesity), but it is prevalently the latter component that positively interferes with bone density [8]. 

The more positive effect of an increase in lean mass on bone compared to fat mass is also demonstrated by the study by Santos et al. [28], which shows that there is a direct relationship between increase in lean mass and bone density (total bone density, femur, and spine) while, on the contrary, sarcopenic obesity causes osteoporosis. An increase in lean mass (muscle strain) also favors an increase of BMD and an improvement in geometry and bone modeling at the level of the upper limbs [51], although, of course, the effect of weight bearing on bone is mainly exerted on the lower limbs. In fact, from this point of view, the literature is controversial. From some studies, it appears that a higher BMI determines a higher BMC (bone mineral content) at the level of the femoral neck and lumbar spine, but not the radius [27,29,34,52]. Some authors observed an increased bone mass at the lumbar spine, radius, and tibia in obese women, but not in men [53,54]. Other authors report a positive correlation between BMI and BMD also in the radial shaft and ultra-distal radius [55]. Quantitative imaging methods, such as HR-pQCT, demonstrated higher cortical BMD, higher trabecular BMD, and greater trabecular number at the distal radius and distal tibia in obese people, but no difference in bone size between obese and normal adults [34,56].

In addition to the mechanical factor, the increase in BMD that has been found in obesity also seems to be linked to the action of estrogens. It is widely demonstrated that estrogens have an important effect on bone metabolism, stimulating bone formation and reducing its resorption [57,58]. There is a close relationship between adipose tissue and estrogen metabolism. Indeed, adipose tissue is one of the major sources of aromatase, which synthesizes estrogens. A higher serum concentration of estrogens was found in obese postmenopausal than in non-obese women [59], and higher 17β-estradiol levels were found in the obese subjects [60].

## 4. Obesity and Fractures

Even though a wide literature shows that the increase in BMI leads to a higher BMD, this increase is not protective against the risk of fractures [18]. This phenomenon has been called “obesity paradox” [19].

Two different factors that make the obese subject more susceptible to fractures must be distinguished: the first is the increase of bone fragility caused by adiposity, the second is the higher risk of falling. 

### 4.1. Increased Bone Fragility

#### 4.1.1. Metabolic Association

Obesity is a condition of chronic dysfunction characterized by a low-grade, systemic inflammatory state. This pathological condition predisposes to the onset of some diseases, such as diabetes, hyperlipidemia, and hypertension. MetS (metabolic syndrome) is mainly characterized by obesity, hyperglycemia, hyperlipidemia, and hypertension, even if its definition has undergone changes over time [61,62,63]. Adipose tissue must be considered an endocrine organ, which regulates many body functions and has a critical role in energy homeostasis, producing, for example, several biologically active substances, like adipokines. Just as adipose tissue can be considered a real organ that acts on the body’s metabolism, bone tissue can also represent an organ that exerts an action on many other organs. The two tissues interact with each other. In order to understand how adipose tissue acts on the musculoskeletal system, we will list a series of substances, in part definable hormones, in part pro-inflammatory cytokines, which have been described in the literature as responsible for the deleterious effect of obesity on bone.

Leptin: Leptin is secreted by the white adipose tissue. Hyperleptinemia found in the obese subject seems to be one of the causes of bone weakening. In fact, leptin has a dual effect on bone. One of these effects is positive: in vitro, leptin stimulates stromal cells to differentiate into osteoblasts, stimulates the proliferation of the latter and inhibits the formation of osteoclasts [64,65,66]. It has also been shown that a knockout of the leptin gene causes a reduction in BMD and bone volume [67]. The negative effect seems to prevail over the positive one [68,69]. This negative effect would be exerted via the central nervous system. Leptin would cause decreased production of serotonin in the hypothalamic neurons, resulting in decreased bone formation [70,71]. In mice lacking leptin or leptin receptors, several authors found a decrease of femur bone mass and an increase of femur bone marrow fat [70,71]. Jansson et al. [72] based on their animal study in rats and mice, hypothesize that, in addition to the action of Leptin, which would have the ability to reduce body weight, also through a reduced food intake [73,74] there would be a homeostat, called “gravitostat” by the authors, located in the weight-bearing lower extremities, which would be activated with the increase in body weight, producing a decrease in fat mass regardless of leptin. The same group of authors, in a more recent paper [75] found that the gravitostat regulates fat mass in obese mice, while leptin regulates fat mass in lean mice, concluding that the gravitostat protects against obesity, whereas undernutrition induces low levels of leptin, with subsequent weight gain. The findings of these two studies lead to an interesting conclusion: obesity has an effect on bone, but bone also has an effect on body weight.

Adiponectine: Adiponectine, secreted by white adipose tissue, is an adipokine that has been proven to stimulate bone formation. It has been shown that adiponectin stimulates osteoblastic proliferation, with an increase in the activity of alkaline phosphatase, and the formation of type I collagen and osteocalcin, all markers of differentiation and maturation of osteoblasts. The osteogenesis of mesenchymal stem cells stimulated by adiponectin is mediated by the adipoR1 phosphorylation of P38 MAPK, which enhances COX-2 (cyclooxygenase2) and BMP2 expression (bone morphogenetic protein 2), a cytokine with considerable osteogenic potential [76,77,78]. In obesity, a low concentration of adiponectin is usually present [79]. This condition induces the reduction in osteoblastogenesis and the increase in osteoclastogenesis [80], through the mechanisms described above and, overall, through a mechanism mediated by inflammation markers. The chronic inflammatory condition present in obesity is likely to be partly linked to the lack of adiponectin. Adiponectin deficiency is also found in insulin-resistant diabetes [79]. This could be one of the links between obesity and diabetes. 

The concentration of adiponectin is inversely proportional to that of numerous inflammatory cytokines, such as C-reactive protein (CRP), IL-6, and TNF-α. It is therefore presumable that the chronic inflammatory state present in obesity expresses a high concentration of these inflammation markers, which are potent inhibitors of adiponectin expression [81]. 

TNF-α: As already mentioned, in the obese subject there is a greater expression of TNF-α (GK57). TNF-α, through multiple mechanisms, leads to an increase in RANKL (RANK-Ligand) [82,83,84]. The latter promotes an osteoclastic bone resorption process. TNF-α also stimulates the production of osteoprotogerin [85].

IL-6: Just as in the case of TNF-α, obesity and insulin resistance cause an increase in interleukin 6 (IL-6) [86], through its overproduction by adipocytes and fibroblasts. IL-6, like TNF-α, also induces osteoclastogenesis and bone resorption [87,88,89].

Resistin: Resistin, a hormone of protein origin produced by visceral adipocytes and macrophages, has a controversial effect on bone. If it seems to favor the proliferation of osteoblasts, it also seems to favor osteoclastic proliferation and the release of inflammatory cytokines [90]. A high concentration of resistin is found in obese people [91].

Peroxisome proliferator-activated receptor gamma (PPARg): According to some authors, the peroxisome proliferator-activated receptor gamma (PPARg), together with its agonists, the thiazolidinediones, can act in the obese subject producing a negative effect on the bone. In fact, PPARgs have the property of promoting the differentiation of mesenchymal cells into adipocytes and blocking the transformation of mesenchymal cells into osteoblasts [92].

Lipid metabolism: The review by Kim et al. [93] carefully examines how the alterations in lipid metabolism present in the obese subject can negatively affect bone metabolism. The lipid alterations to which such consequences can be attributed are different and complex, with the involvement of SREBP, cholesterol, LXRs and RXRs, fatty acids, statins. The latter can impact the phenotype of osteoclasts and osteoblasts in pathological conditions.

Vitamin D: Vitamin D deficiency causes a reduction of calcium resorption and, consequently, osteoporosis and osteomalacia. In the obese, the serum levels of vitamin D are significantly lower than in non-obese [94,95,96]. However, as we have previously reported, the obese patient’s BMD is higher. The incongruity of this phenomenon can be explained by the fact that a wide amount of vitamin D in the obese is stored in the largely represented adipose tissue, causing a serum hypovitaminosis D. This hypovitaminosis is only apparent though, since vitamin D in the adipose deposits is always available and, therefore, the obese subjects are not affected by the negative effects of the deficiency of this vitamin [97]. In obese patients, it is very common (43% of the morbidly obese adults) to observe a secondary hyperparathyroidism, that can negatively impact skeletal health [98,99].

Peptide YY: Not strictly related to metabolism, is the level of Peptide YY. Although it must be confirmed, the role of this peptide seems to have an influence both on obesity and bone mass. Peptide YY promotes satiety. PYY-deficient mice (Pyy(−/−)) have osteopenia with a reduction in trabecular bone mass and a deficit in bone strength. PYY levels are lower in obese adults and the elevation of PYY seen after a meal in lean subjects is blunted in obesity [100,101]. 

The positive and negative effects of the various substances in the obese patient are schematized in Table 1.

#### 4.1.2. Fat Bone Marrow

Bone marrow is an important deposit of fat, at the level of the “yellow” areas. Marrow adipose tissue (BAT) is estimated to occupy 70% of the marrow space by adulthood [102] and accounts for about 8% of total fat mass [103]. In the obese subject, bone marrow fat fraction (BMFF) was shown to be higher than in the normal weight subject [104]. The adipocytes of the bone marrow are responsible for the secretion of adipokines, some of which induce the release of various inflammatory cytokines mentioned above, such as TNFα and IL-6. 

The correlation between obesity and bone fragility can also originate from the adipose bone marrow, which has been shown to interfere with bone metabolism. It is not a coincidence that BMFF increases in obesity, in old age and in osteoporosis, especially in postmenopausal women [105]. Actually, adipocytes and osteoblasts have a common origin, which are pluripotential, bone marrow-derived mesenchymal stem cells [106]. It cannot be ruled out that the inability of the latter cells to differentiate into osteoblasts leads to an increased differentiation into adipocytes. The literature widely demonstrates that the presence of a greater amount of fat in the bone marrow induces osteoporosis [107,108,109,110,111,112]. Indeed, an increase in marrow fat content has been demonstrated in obese women with low BMD, and Wehrli et al. reported that bone marrow adipose tissue in the lumbar spine is an independent predictive factor of fracture [104,113,114,115]. 

It is also important to mention the importance of palmitate, reported by some authors [116]. According to these authors, the lipotoxic effect of BAT is mainly due to the action of palmitate, which would have its toxic effect, especially on bone cells, mainly osteoblasts.

#### 4.1.3. Genetic Predisposition

In some subjects, obesity is dependent on a mutation of the *FTO*
*(FaT mass and Obesity-associated protein) gene* [117]. It has been demonstrated that the deletion of FTO in mice leads to increased death of osteoblasts and bone loss [118]. It could be inferred that subjects carrying the *FTO gene* mutation are more predisposed to osteoporosis due to the depletion of osteoblasts.

### 4.2. Type of Adiposity

Abdominal fat is composed of abdominal subcutaneous fat and intraabdominal fat. Intraabdominal adipose tissue is composed of visceral, or intraperitoneal, fat [117] and its accumulation is the cause of central adiposity. Abdominal obesity is an index of visceral (or central) adiposity and can be measured through the waist circumference. It must be distinguished from obesity in general, which is measured with BMI. From the literature it emerges that, more than obesity, evaluable with BMI, it is the excess of visceral adiposity that induces damage of bone microstructure. Visceral adipose tissue (VAT) has been associated with lower trabecular bone volume, lower bone formation rate, lower stiffness, and higher cortical porosity [119]. Numerous papers published in the literature highlight that VAT is an independent negative determining factor of bone density in obesity [104,120,121,122]. A Korean study on a large population of postmenopausal women (*n* = 3058) found that the prevalence of osteoporosis in women with waist circumference (WC) obesity (>80 cm) was higher than in women with BMI obesity (>25 kg/m^2^) [123]. A study by Cao et al. showed low levels of IGF-1 in VAT. IGF-1 has an anabolic action on osteoblasts [124]. Some authors report that an increase in VAT causes a higher release of pro-inflammatory molecules such as TNF and IL-6, which are very important in the genesis of osteoporosis [125,126].

It is interesting what some authors report, arguing that visceral fat has an adverse effect on bone mass, while subcutaneous fat would have beneficial effects [122].

### 4.3. Age and Sex

It is not easy to draw conclusions regarding the differences in bone quantity and quality and the risk of fracture in obese subjects based on age and sex. It is well known that postmenopausal women have a much higher risk of osteoporosis than premenopausal women and men, although there are numerous studies that conclude that osteoporosis in men should not be underestimated [127]. As regards the influence of age and sex in the relationship between obesity and bone fragility, biasing factors are unavoidable in the assessment of fracture risk: the risk of osteoporosis and fractures increases with age, independently from body weight, and, in menopause, the abrupt decrease of estrogens is the main cause of osteoporosis. 

In the literature, it is possible to find conflicting data on the responsibility of obesity on bone health based on age and sex. Several studies conducted by Asian authors deal with metabolic syndrome in general. In two retrospective studies on populations of over 50, it appears that MetS has a protective effect on bone in men but not in women [128,129]. These conclusions are confirmed by the studies of Eckstein et al. and Hernandez et al. [130,131]. Totally opposite results emerge from some studies carried out on the Korean population [58,132,133]. Zhou et al., in their meta-analysis in which nine studies were selected for a total population of 18,380, confirm that MetS has more harmful effects on bone in men than in women [134]. Even considering the “menopause” factor influencing the results, it appears that obesity determines a significantly higher BMD increase in postmenopausal women than in premenopausal women [21,30,34].

### 4.4. Obese’s Fracture Site Paradox

As we have seen, the literature demonstrates that the risk of fracture in the obese is increased. In fact, the published scientific data highlight that the increased risk does not concern all fractures, but only some sites are affected by a higher incidence of fractures. This site-specificity of fracture risk could be called “obese’s fracture site paradox” (in analogy with the “obesity paradox” mentioned above), since the increased risk of fracture paradoxically does not concern the typical sites of osteoporotic fractures [135,136] but other, less common, sites. Specifically, most of the available evidence supports a lower risk of hip, vertebral, and wrist fractures in obese adults [137,138], whereas a higher risk of ankle, upper leg, and humerus fractures has been found [139,140,141,142]. In a study carried out by Compston et al. on 3628 fractures in 52,939 post-menopausal women followed for 3 years, the authors concluded that BMI was protective for hip and wrist, whereas the risk of ankle fractures increased (HR 1.05 [1.02–1.07]) (*p* < 0.01) [139]. 

One wonders what the reasons for this anomalous distribution of these fractures may be. The hypotheses that can be formulated are essentially related to the mechanism of falling. A first factor could be the hip padding, that is, the presence of an abundant fat pad around the pelvis that could protect obese individuals from hip fractures, while at the level of the legs and upper limb the protection by fat is minimal [143,144,145]. Another factor could be how obese people fall, since they are more prone to fall backward or sideward [146]. A further hypothesized factor is the tendency of the obese to excessive introversion and extroversion of the ankle and lower leg. That would predispose to sprains and fractures of the ankle [47].

### 4.5. High-Fat Diet

In order to better understand the relationship between obesity and bone health, the analysis of the numerous studies on the effects of high-fat diet (HFD) can be interesting. Most of the findings on the effects of high fat diet come from animal studies.

Ionova-Martin et al. report that obesity induced in C57BL/6 mice by HFD is associated with an increased bone quantity (larger bone size and mineral content), but also with a decrease in bone quality, as evidenced by lower size-independent mechanical properties [147]. Other demonstrations of the deleterious effect of HFD in mice are reported by Fujita et al., who showed the reduction of trabecular bone density and by Patch et al., which reported an increase in bone resorption [148,149]. Several other studies have confirmed the decreased bone mass in mice following HFD [149,150,151,152,153,154,155], although some studies in rats reported opposite results [156,157,158,159]. However, most of the studies report harmful effects of high-fat diet on bone. It has been shown that the component of the bone tissue that suffers mostly from this type of dietary intake is the cancellous bone. Several authors found that the consequences of HFD are the decrease of bone trabecular density [148] and of bone trabecular volume fraction, bone mineral content, and quantity [154]. The studies on cortical bone do not have comparable evidence [124,160,161]. This can be explained by the fact that cancellous bone is more sensitive to bone turnover, as it has a greater remodeling action, likely due to its larger surface to volume ratio [162]. Another reason may be the fact that cortical bone is less affected by bone resorption because the body weight load mainly acts on the cortical bone, strengthening it. HFD, in addition to exerting a deleterious action on bone structure, has also a harmful effect on the cellular component: it induces osteoclast hyperactivity and bone resorption [163] mainly through the RANKL/RANK/OPG signaling pathway. Shu et al. (2015) found an increased osteoclast number in the femoral metaphyseal sections of HFD-fed mice, associated with the finding of RANKL, TNF, and PPARγ in bone marrow cells [153]. In fact, an increase in osteoblast function was also noted in the same study. To explain these findings, it can be hypothesized that, despite the harmful effect of HFD on bone by changing the bone marrow microenvironment, the weight gain of animals gives the bone a greater biomechanical stimulus, which only partially reduces bone fragility.

### 4.6. Gut Microbiota

Microbiome science is relatively new and evolving. Gut microbiota (GM) dysbiosis has been identified in various diseases, such as hypertension, Alzheimer disease, type 2 diabetes, depression, and also in obesity [164,165,166,167,168,169]. Several studies have also shown a relationship of GM dysbiosis with bone health [170]. It can be inferred that the relationship between obesity and bone can also be conditioned by GM. In an animal study by Wang et al. [171,172], the GM dysbiosis induced by the transfer of feces from osteoporotic senile rats to young rats, made the latter osteoporotic. From the study by Zhou et al. on 264 obese or overweight subjects [173], it emerges that the gut microbiota-related metabolite trimethylamine N-oxide (TMAO) protects against BMD reduction during weight loss. An important study on this topic is that of Fernández-Murga et al. [172].

The experiment was conducted on two groups of mice: both groups of mice were fed a high-fat diet (HFD) for 14 days, but in one of the two groups the diet was supplemented with *Bifidobacterium pseudocatenulatum CECT 7765*. In HFD-fed mice, bone alterations were detected, such as reduced volumetric bone mineral density in the trabecular bone and deteriorated trabecular architecture in bone volumetric fraction, trabecular number, and trabecular pattern factor at the level of the distal femur. In HFD-fed mice supplemented with *B. pseudocatenulatum CECT 7765*, the findings were the following: low negative effect on bone microstructure, increased Wnt/β-catenin pathway gene expression (which improves BMD), and decreased serum C-terminal telopeptide (CTX) and parathormone. These findings demonstrate the protective effect of *B. pseudocatenulatum CECT 7765* on bone in obese mice.

### 4.7. Effect of Weight Loss

From what has been said so far, it can be inferred that the increase of body weight generally induces an increase of BMD, but that the bone is not protected by it, on the contrary, its microstructure is damaged and the risk of fracture increases. What happens when the obese subject loses weight? It is interesting to analyze the studies that deal with this topic. 

In rats, long-term calorie restriction with subsequent decrease in body weight is associated with reduced bone mass [7]. All the studies we found in the literature highlight how weight loss, both intentional and unintentional, leads to a loss of BMD at the hip and proximal humerus [174,175,176], with a consequent increase of the risk of fracture at these sites, while this effect is not detected at the spine [177,178,179,180]. The results of the study by Ensrud et al. showed that older women had a 35% decline in hip BMD for every 5 kg lost, compared to weight-stable women, and doubled hip fracture risk. A human study showed similar but less striking results: the adjusted average rate of change in total hip BMD was 0.1%/year in men who gained weight, −0.3%/year in weight-stable men, while men who lost weight had a decrease in BMD of 1.4%/year. [174,175,176]. Similar results are reported by other RCTs [168,170,172,173]. If diet-induced weight loss is combined with exercise training, there is an attenuation of the loss of total hip BMD in older obese patients [168,169,173]. One of the hypotheses that would explain the negative effect of weight loss on BMD is the decreased intestinal absorption of calcium in a manner independent from the effects of vitamin D [172], but this theory needs to be better verified. A study by Shah et al. [173] on 107 older (age > 65 years) obese subjects (body mass index (BMI) ≥ 30 kg/m^2^), compares the following four groups of individuals: a diet–exercise group (a group that had an exercise program associated with the diet), a diet group, an exercise group, and a control group, for 1 year. Body weight decreased in the diet (−9.6%) and diet–exercise (−9.4%) groups, but, regardless of weight loss (which was comparable), the comparison between these two groups showed, in the diet group (no exercises), a significantly greater decline of hip BMD (−2.6% versus 1.1%), and a serum C-terminal telopeptide (CTX) and osteocalcin concentration increase (31% and 24%, respectively). Serum leptin and estradiol concentrations decreased in both groups. What could be inferred from this study is that subjects who practice physical activity have less bone loss because of the lower loss of lean mass. In fact, the authors report that changes in lean body mass were independent predictors of changes in hip BMD. Then, the explanation may be that, in subjects who perform exercises, repeated muscle loading reduces the damaging effect of weight loss on bone.

### 4.8. Bariatric Surgery

Strictly connected with the effect of weight loss are the consequences of bariatric surgery, on which numerous papers have been published. 

The most performed surgical techniques in bariatric surgery are the following: laparoscopic adjustable gastric banding (LAGB), sleeve gastrectomy (SL), roux-en-Y gastric bypass (RYGB), biliopancreatic diversion with duodenal switch (BPD/DS) [181].

Related to this topic, available evidence suggests the following conclusions about the consequences of bariatric surgery: (1) decrease in BMD and areal BMD (aBMD) [13,182,183,184,185,186,187,188,189,190], with endocortical resorption, evidenced by the decrease in the number of trabeculae and a great increase of cortical porosity [12,13,186]; (2) early and dramatic increase of biochemical markers of bone turnover, such as serum C-terminal telopeptide (CTx), especially after RYGB [183,184,190,191,192]; (3) as far as the type of surgery is concerned, these consequences seem to be rare after LAGB [193], while they are frequent after RYGB and BPD-DS [194,195,196,197]; (4) the risk of fracture increases, but the most frequent fracture sites are different from those typically found in the obese subjects. In detail, the mostly incident fractures reported are wrist, humerus, spine, hip, femur [194,195,196], clavicle, scapula, sternum, foot [197]; (5) The risk of fracture increases at a longtime distance from the operation (it starts to increase between 2 and 5 years after surgery) [194,195,196,197]; (6) The negative consequences on bone can be mitigated with exercise [198].

It should be kept in mind that most of the patients who undergo bariatric surgery are women between the ages of 30 and 40 [163]. It follows that the risk of fracture in that age group is low, but it is, however, increased after this type of surgery.

There are several hypotheses on the etiopathogenesis of the negative influence of bariatric surgery on bone. The etiopathogenesis is presumably multifactorial, and mechanisms may involve nutritional factors, mechanical unloading, hormonal factors, and changes in body composition and bone marrow fat. The first indisputable factor is hypovitaminosis D, which occurs in the operated subject. Already before surgery, as previously described, the patient is probably deficient in vitamin D, because of their obesity, but in the postoperative period, this deficiency worsens and, unfortunately, the therapeutic intake of vitamin is unable to compensate for the deficit, with persistence of a serum concentration of 25OHD below 30 ng/mL [199]. As a result, there is severe calcium malabsorption, which is reduced by up to 7% after RYGB by 6 months from surgery [190]. Another consequence is that subjects undergoing bariatric surgery almost always have high levels of parathyroid hormone [200,201]. Another conceivable cause is that the patient’s sudden weight loss can significantly affect the bone stoke as a result of mechanical unloading. This last factor can contribute to the increment of bone fragility, but it cannot be considered the most important, also because it would not explain the increase in fractures in sites not subjected to load, such as the upper limb.

In patients submitted to bariatric surgery, several hormonal changes have been detected. In particular, the increase of adiponectin and peptide YY and the reduction in estradiol, leptin, insulin, and ghrelin (the latter not always increases) causes a decrease of bone mass, while the increase in testosterone, GLP-1, and IGF-1 is able to induce bone gain [188,189,202,203].

Another hypothesized mechanism to explain bone loss after bariatric surgery is the reduction of lean mass, which would lead to a decline in aBMD [13,189,204,205].

### 4.9. Osteosarcopenic Obesity Syndrome

“Sarcopenic obesity” is characterized by loss of muscle mass due to obesity [206,207,208]. The combination of osteoporosis/osteopenia and sarcopenia has been called “osteosarcopenia” [209,210,211]. The “osteosarcopenic obesity syndrome”, first described by Ilich et al. [212], is a syndrome characterized by the combination of three conditions: adiposity, with infiltration of fat in muscle tissue and bone, sarcopenia, and osteopenia/osteoporosis. The condition that most of all induces the other two is obesity, as a form of low chronic inflammation that causes the release of numerous cytokines harmful to the bone and muscle and produces fatty infiltration of the muscles, making the latter less strong and efficient [212,213,214]. In sarcopenia, it has also been observed the release of specific muscle cytokines, such as myostatin, which can inhibit osteogenic differentiation of BMSCs, as well as osteoblast differentiation and mineralization [214,215,216]. 

Sarcopenic obesity is a highly prevalent condition in the elderly. According to data from the European Working Group on Sarcopenia in Older People (EWGSOP), the prevalence of sarcopenia increases by 11% in subjects between 50 and 59 years of age and by 37% in subjects aged 80 and over [217]. Sarcopenic obesity sums up the effects of obesity and sarcopenia on elevation of fracture risk. Scott et al. report that older adults with sarcopenic obesity have a three times greater risk of fracture than older adults with nonsarcopenic obesity and controls (no sarcopenia, no obesity) [218]. It has also been shown that older adults with sarcopenic obesity have higher percentage of nonvertebral fractures, compared with those with sarcopenia alone and those with obesity alone [218,219].

The reasons for this increased risk of fractures are the following: (1) the decreased bone strength, partly due to the age-related decrease of BMD [220,221] and, as we said earlier, to the action of myostatin [215,216] and to the worse bone quality connected with obesity [14,15,16,17,18]; (2) the greater risk of falling [213,222,223,224,225] connected to the muscular weakness of sarcopenia, postural instability and reduced physical activity [224] derived both from the condition of obesity and from that of sarcopenia [226] without forgetting that we are generally speaking of elderly patients, whose physical abilities are naturally reduced. The greater incidence of falls in the subject affected by sarcopenic obesity is reported by several authors [213,222,223,224,225]. 

### 4.10. Obesity and Falls

One of the causes of the increased risk of fractures in the obese subject is the greater tendency to fall compared to the non-obese. 

The increased risk of falling in obese people is reported by various authors [223,227,228,229,230]. In a meta-analysis published in 2019, which analyzed 31 observational studies [231], it emerges that obese people not only have a greater risk of falling than non-obese, but also that obese subjects have a significantly greater tendency to experience multiple falls.

Several causes for the greater risk of falling in obese people are reported in the literature: (1) excessive body weight reduces the subject’s agility and therefore his ability to move skillfully avoiding obstacles [232] and slows down the reaction time in supporting the body mass during falling [233,234,235]. (2) Postural instability. The body center stability is lower in obese people [229,230], especially in older women with central adiposity [236]. In a cross-sectional study conducted on 201 older adults, Azevedo-Garcia et al. [237] concluded that obesity is associated with postural balance on unstable surfaces. The lower stability was also associated with the greater pressure on the heels exerted by the obese subject, which decompensates the load distribution and alters the correct dynamics of the gait [238]. (3) The poorer physical activity performed on average by obese subjects [239,240] also influences their lower balance capacity, both because of the lower muscle strength and the lower agility related to the poor daily exercise [241]. (4) The term “dynapenic obesity” is referred to the association of obesity with lower muscle strength, due to fatty infiltration of the muscles [219,241,242,243]. This leads to a greater predisposition to falls. Dynapenic obesity is also present in “sarcopenic obesity”, a highly prevalent condition in the elderly which, as already explained in a previous paragraph, involves a greater risk of falls and fractures [213,222,223,224,225]. (5) Obesity is associated with some diseases, of which it can be cause or effect, such as diabetes, cardiovascular diseases, chronic pulmonary diseases, sleep apnea, hypertension [232]. These conditions can be associated to peripheral neuropathy, orthostatic hypotension, general weakness, all predisposing to falls [244]. We should not forget that hip and knee osteoarthritis is a disease to which overweight people are predisposed [245,246]. The arthritic subject is often affected by limping or, in any case, by pain that makes their gait less fluid, less stable, predisposing them to a greater risk of falling [247,248,249].

In Figure 1 are listed the possible causes for the increased risk of falls in obese people and the hypotheses for the paradoxical fracture site distribution.

## 5. Obesity and Bone Health in Children and Adolescents

Childhood obesity can nowadays be seen as an international public health problem with epidemic proportions. The most recent epidemiological data have revealed a strong increase of childhood overweight and obesity [250]. This has consequently caused a significant public health burden with high costs and resources utilization. The changes in metabolism and bone structure in obese child and adolescent can cause consequences during childhood itself, but also consequences in later stages of life with the development of chronic diseases and fractures [251,252]. 

A significant amount of bone mineral content finally acquired by adult individuals depends on processes occurring during the period of puberty. During childhood and adolescence, the rate of bone formation exceeds that of bone resorption, favouring bone acquisition. Almost half of the adult bone mass is acquired during adolescence. In both genders, peak bone mass acquisition occurs around the seventh or eighth month after the maximum longitudinal bone growth (growth spurt) as a result of the high concentrations of hormones [253,254,255].

Impaired bone growth during childhood and adolescence is thought to be able to lead to suboptimal peak bone mass and to increase the risk of developing osteopenia/osteoporosis and fractures in old age. In fact, adolescence is considered a critical period for bone mass gain. The greatest gain in bone occurs during this phase, when peak bone mass is reached [256].

It has been shown that up to 80% of peak bone mass has a significant genetic influence, whilst the remaining 20% is thought to be depending on environmental factors, which have the potential to cause a reduced bone mineral density and increased risk of fracture. Therefore, it appears evident that factors affecting the bone mass during childhood can determine a reduction in bone mass in the adulthood period. There is still ongoing discussion on whether there is a positive or negative effect of fat mass on bone, or even neutral effect, and whether excessive adiposity is either beneficial or detrimental to the growing skeleton [257]. 

As in adulthood, the positive effect of being overweight on BMD also occurs in children [257,258]. This early increase in bone mass may lead to an accelerated skeletal maturity and advanced bone age beyond the actual child’s age [259,260]. Oh et al., in a study of 232 children between the ages of 6 and 15, found that an increase in weight, height, BMI, and waist circumference percentiles all favored advanced bone age [261]. One could infer that an acceleration of bone development could create an anomaly of its structure which would also influence bone quality in adulthood.

In children, there can be a risk of overestimation or underestimation of the true bone density, respectively, for taller children with larger bones and shorter children with smaller bones. Several studies have shown that bone mass is reduced in obese children, but many debates still exist. Neither the use of DEXA Scan has helped the researchers in achieving a consensus on the topic. On the other hand, BMC is recommended as the best bone parameter to assess bone mass status in children and adolescents [257].

Exactly as in adults, the phenomenon of “obesity paradox” is observed in children. Even in children, overweight has a positive effect on BMD, but the incidence of fractures is higher than in non-obese individuals [252]. An interesting large study conducted in Catalonia by Lane et al. [262] found that preschool obesity is associated with an increase in fracture risk in teenage children. Studies related to slipped capital femoral epiphyses and tibia vara helped in understanding and highlighting that skeletal complications might be caused by excessive mechanical loading due to excessive adipose tissue [252,263].

Another interesting, proven aspect is that related to the different and opposing effects of total adiposity and central adiposity on body bone mass (similarly to the adult population). Whereas total body fat had a positive association with bone mass, visceral fat had a negative relationship with bone mass [264]. This concept has been demonstrated even in adults, as reported in the previous paragraphs. In children and adults with high level of visceral adiposity, physiological secretion of growth hormone (GH) is impaired. This in turn may impact on bone mass accrual and skeletal integrity as GH promotes myogenesis and osteoblastogenesis and regulates the hepatic generation of insulin-like growth factor 1 (IGF-1) which promotes chondrogenesis at the growth plate and osteoblast proliferation and activity. IGF-1 also acts indirectly to promote renal tubular resorption of phosphate and the synthesis of calcitriol. Impaired GH secretion may be compounded by highly caloric low protein diets in obese children that impacts on the synthesis of IGF-1. Androgens stimulate the differentiation and proliferation of osteoblasts via androgen receptors, decrease osteoblast and osteocyte apoptosis, and indirectly and directly modify osteoclastogenesis in favour of a reduction in bone resorption. Indirectly, androgens upregulate the Transforming Growth Factor β (TGF-β) and Insulin-like Growth Factors (IGFs), promoting bone formation, and downregulate Interleukin 6 (IL-6), thus inhibiting osteoclastogenesis [252].

It is very likely that a “melting pot” of factors interplay altogether with increased childhood fracture rates. These might involve genetic, hormonal, environmental, and behavioural factors, such as inadequate calcium intake, low vitamin D levels, inadequate physical activity, weight, diet. These predisposing factors need to be precipitated by further events able to trigger the underlying predisposition. The long-term effect is an increased risk for fractures and the overall health status [251]. In fact, with regards to the hypotheses that could justify the increased risk of fractures in obese children, the discussion is still open. On the one hand, it can be attributed to poor bone quality, for reasons that can also be found in adults, i.e., the metabolic syndrome with its low inflammation process (and the consequent release of TNFα, IL-6 and other cytokines), the excessive production of some adipokines, the excessive release of estrogens by the overabundant adipose tissue, and all the other metabolic causes already described in the previous paragraphs [256]. Most factors and substances (together with their effects) with a significant impact on bone metabolism are shared between the adult and the paediatric population (leptin, adiponectin, osteocalcin, activation of the immune system, chronic low-grade inflammation with proinflammatory cytokines, vitamin D, parathyroid hormone, etc.) [264,265,266,267,268,269].

A studied hypothesis is the existence of a strong relationship between type 2 diabetes and osteoporotic fractures. In fact, insulin resistance can be seen as the linking factor between poor bone health and childhood obesity. Another related hypothesis is that childhood obesity could promotes low bone mass accrual and risk for diabetes through several studied mechanisms [265]. 

On the other hand, it is thought that the increase in fracture risk simply results from the increased propensity to falls seen in obese children due to changes in postural stability and gait. However, this could well be caused by a suboptimal response of skeletal adaption to body size resulting in a mismatch between body and bone size, thus increasing fracture risk [257]. It is also supported the hypothesis that the increased bone mineral density in obese adolescents may not be sufficient to overcome the significant greater forces that are generated when an overweight child falls [270]. 

Very interesting is the topic related to the anatomical distribution of fractures. If the “obese’s fracture site paradox”, previously described, corresponds to a higher incidence of fractures in the obese at uncommon sites in comparison to non-obese subjects, a comparable fact occurs in children, where the distribution of fractures in the obese child is different from that of the non-obese child.

The fractures that have a higher incidence in paediatric and adolescent age are those of the distal forearm. According to a study by Naranje et al. [271], conducted on a large American population of individuals between 10 and 19 years of age, forearm fractures accounted for 17.8% of all fractures, whereas finger and wrist fractures were the second and third most common, respectively. Finger and hand fractures were most common for age groups 10 to 14 and 15 to 19 years, respectively. A Swedish study conducted in a population of youths < or =19 years of age [272] showed that the first most common fracture site was the forearm, followed by the clavicle and the fingers.

Obese children are significantly more likely to sustain lower extremity injuries than upper extremity injuries and less likely to sustain head and face injuries than non-obese children. Pre-school obese children have been reported to have an increased incidence of both upper and lower limb fractures in childhood compared with contemporaries of normal weigh [270,273]. The fracture pattern of both upper and lower limbs differs between the obese children and the normal-weight children. An interesting study by Nhan et al. [274] divides the study population (608 patients) into three groups: 58% normal weight, 23% overweight, and 19% obese children. Overweight/obese patients sustained significantly more upper-extremity physeal fractures and greater proportions of complete fractures compared with normal-weight children. 

We can therefore summarize that obese children have a greater risk of fracture than normal weight children, both of the upper and lower limbs, that the ratio lower limb/upper limb fractures is greater and that the fracture pattern also differs according to body weight.

In Table 2 the characteristics common to the obese child and the obese adults, as well as the peculiarities of the paediatric population are illustrated.

## 6. Conclusions

For many years it was thought that a high body weight was a protective factor against osteoporosis. In fact, it is true that, in heavier subjects, DEXA shows an increase in BMD, but the literature agrees that the obese subject is at greater risk of fractures than a normal-weight individual. The greater risk of fractures is due to numerous factors, which can be grouped into two large groups: metabolic factors and increased risk of falls. Fractures of the obese have an atypical distribution, both in adulthood and in childhood. Obesity is therefore confirmed as a very dangerous condition for men and women, due to the possible serious consequences on numerous systems, including the skeleton.

## Figures and Tables

**Figure 1 ijms-22-13662-f001:**
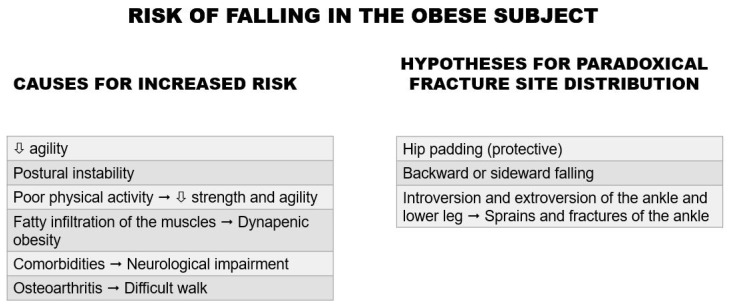
Risk of falling in the obese subject. In the table on the left, the possible causes for the increased risks of falling in the obese subject are listed. In the table on the right, the hypotheses for the paradoxical fracture location are illustrated.

**Table 1 ijms-22-13662-t001:** Schematic representation of the positive and negative effects of various substances in the obese patient. As can be seen, the increase in Leptin can have both a positive and a negative effect, although it is the latter that prevails. TNFα and IL-6 are produced in excess due to the dysmetabolic action of obesity, but also adiponectin, as indicated by the arrow, leads to an increase of these inflammatory cytokines.

Effects of Obesity on Bone
Positive Effects	Negative Effects ( 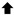 Fractures)
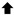 mechanical load	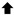 leptin
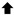 estrogens	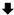 adiponectin
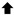 leptin	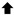 TNF-α
	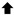 IL-6
	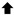 PPARg
	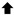 PTH
	Dyslipidemia
	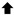 palmitate

**Table 2 ijms-22-13662-t002:** Aspects and consequences of obesity in relation to bone health that are shared among the adult and paediatric populations (left column) and those peculiar of the paediatric population (right column).

Aspects Shared by the Adult and Paediatric Population	Specific Aspects of the Paediatric Population
Increase of overweight and obesity	A significant amount of bone mass and mineral content in adults depends on processes occurring during puberty
Development of chronic diseases and fractures	Risk of overestimation or underestimation of the true bone density
Positive effect of overweight on BMD	Increased risk of fractures in obese children
Phenomenon of “obesity paradox”	Strong relationship between type 2 diabetes and osteoporotic fractures
Higher incidence of fractures than in non-obese individuals	Increased propensity to falls due to changes in postural stability and gait
Positive association of total body fat and bone mass	“Obese’s fracture site paradox”: greater incidence of fractures in the obese at uncommon sites in comparison to non-obese subjects
Negative relationship of visceral fat and bone mass	Significantly higher risk to sustain lower extremity injuries than upper extremity injuries and less likely to sustain head and face injuries
Predisposing factors for increased fracture rates need to be precipitated by further events able to trigger the underlying predisposition	
Negative influence on bone of inflammatory cytokines, adipokines, estrogens, and all other metabolic causes	

## Data Availability

Not applicable.

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
