# Peer review of "Obesity and Bone: A Complex Relationship"

_ijms, 2021, doi:10.3390/ijms222413662_

Round 1
Reviewer 1 Report
Review comments on ijms-1485796: Obesity and bone: a complex relationship
This is a literature review on the relationship between obesity and bone. The authors summarized information from 263 references and structured a narrative review, which finally resulted in a conclusion that high body weight was a protective factor again osteoporosis, but it induced risk of bone fractures. This conclusion is not new because some similar reviews have mentioned it (such as 10.12688/f1000research.20875.1, 10.1007/s00223-016-0229-0, 10.1007/s42000-018-0018-4, 10.3389/fcell.2020.600181). In addition, there are several issues to consider in this manuscript, as listed below. Please modify and improve the manuscript.
- As mentioned, there are some similar reviews in the literature. The authors should emphasize the novelty and significance of this review. In detail, the Introduction section should be expanded to clarify (i) relevant issues of bone, (ii) why identifying the relationship between obesity and bone are essential, and (iii) whether similar reviews are available (if yes, what are the novelty and contribution of this review to the field?).
- Introduction: it is required to include clear definitions of obesity and overweight and the criteria to define them.
- Section 2-Risk of fracture: there were only 2 sentences in this section. It is better to incorporate this section into another section.
- Figure 2 is better to transform into a table.
- Another mechanism for the effects of obesity on bone is that some mutations in the fat mass and obesity-associated (FTO) gene simultaneously result in weight gain and bone fragility (10.12688/f1000research.20875.1).
- The manuscript focused on how body weight affects bone. How about the effects of bone on body weight? Please consider this article (10.1073/pnas.1715687114) and other relevant ones.
- The authors may consider summarizing information of some (sub)sections into tables/figures, which can be more informative (e.g., sections 4 and 6).
Some minor points:
- There are many short paragraphs with 1-3 sentences. They should be better combined with other paragraphs.
- Line 32: BMD should be defined.
- Line 33: “World Health Organization” was abbreviated as WHO in line 7.
- Lines 60-62: “For many years it was thought that the people most at risk of fractures were lean women. The lower BMI was thought to increase the risk of osteoporosis, while the higher body weight to give protection against fractures.” Please include references.
- Line 75: please include the missing information “while [34], when…”
- There was no author list to detect inappropriate self-citations.
Author Response
Please modify and improve the manuscript.
The changes in the manuscript are written in red.
1. As mentioned, there are some similar reviews in the literature. The authors should emphasize the novelty and significance of this review. In detail, the Introduction section should be expanded to clarify (i) relevant issues of bone, (ii) why identifying the relationship between obesity and bone are essential, and (iii) whether similar reviews are available (if yes, what are the novelty and contribution of this review to the field?).
Thank you for your suggestions. Given that it is not an original article and therefore cannot present new scientific evidence, we believe that it is a very complete review, as also highlighted in the introduction in lines 24-26. As suggested, we have expanded the introduction by specifying the importance of the topic.
2. Introduction: it is required to include clear definitions of obesity and overweight and the criteria to define them.
The definition of obesity and overweight are clearly specified in the introduction (lines 6, 7). We added a more detailed definition of overweight (line 7, 8).
3. Section 2-Risk of fracture: there were only 2 sentences in this section. It is better to incorporate this section into another section.
Thank you for the suggestion. We put those two sentences a the end of the introduction.
4. Figure 2 is better to transform into a table.
Thank you. We assumed that the figure was visually more useful as we had schematized it, but, as per your suggestion, we have transformed it into a table (table 1).
5. Another mechanism for the effects of obesity on bone is that some mutations in the fat mass and obesity-associated (FTO) gene simultaneously result in weight gain and bone fragility (10.12688/f1000research.20875.1).
Thank you for your suggestion. We have added a short paragraph (4.3.3, lines 231-235)
6. The manuscript focused on how body weight affects bone. How about the effects of bone on body weight? Please consider this article (10.1073/pnas.1715687114) and other relevant ones.
Although we had already mentioned this topic in a part of the paper (lines 76-82), we have added a specific part in lines 142-150.
7. The authors may consider summarizing information of some (sub)sections into tables/figures, which can be more informative (e.g., sections 4 and 6).
We added table 2 in order to summarize section 5. It is very hard to summarize the remaining sections with tables.
Some minor points:
1. There are many short paragraphs with 1-3 sentences. They should be better combined with other paragraphs.
Please suggest how to combine the different paragraphs. The subdivisions have been made to make the text clearer.
2. Line 32: BMD should be defined.
BMD has been defined in the previous part of the manuscript (line 17)
3. Line 33: “World Health Organization” was abbreviated as WHO in line 7.
It is the correct abbreviation.
4. Lines 60-62: “For many years it was thought that the people most at risk of fractures were lean women. The lower BMI was thought to increase the risk of osteoporosis, while the higher body weight to give protection against fractures.” Please include references.
Thank you for noticing the lack. We have added the references (line 67).
5. Line 75: please include the missing information “while [34], when…”
We are sorry. We made the correction.
6. There was no author list to detect inappropriate self-citations.
We are sorry, we do not understand what you mean. However, the self-citations are references nr 21, 38, 133.
Reviewer 2 Report
Thank you for giving me the opportunity to read and review this interesting manuscript.
It is well written and easy to read, complete and informative.
The focus is analyzed by several point of view, but in my opinion it lacks important informations about the role of our microbiota and probiotics in the pathophysiology of metabolic disorders, including obesity and bone diseases.
I suggest to add such valuable information discussing at least the following articles (PMIDs): 34480200, 31332027, 32727337, 31952249, 32795675, 33255588, 34394379, 33245146, 34061595.
Author Response
Thank you for giving me the opportunity to read and review this interesting manuscript.
It is well written and easy to read, complete and informative.
The focus is analyzed by several point of view, but in my opinion it lacks important informations about the role of our microbiota and probiotics in the pathophysiology of metabolic disorders, including obesity and bone diseases.
I suggest to add such valuable information discussing at least the following articles (PMIDs): 34480200, 31332027, 32727337, 31952249, 32795675, 33255588, 34394379, 33245146, 34061595.
Thank you very much for this suggestion. We added section 4.6. (lines 320-339).
Round 2
Reviewer 1 Report
Review comments on ijms-1485796-R1: Obesity and bone: a complex relationship
The authors appropriately revised the manuscript and solved most of the previous concerns. The manuscript can be accepted for publication in the International Journal of Molecular Sciences after correcting some minor points below.
1. Line 232: the abbreviation FTO should be defined.
2. Line 39: “World Health Organization” should be changed to "WHO" since this abbreviation was previously defined in line 4.
3. Lines 7-8: "BMI greater than or equal to 25" should be changed to "BMI≥25".
Author Response
Thank you,
the corrections have been made as you suggested.